# ON THE BENEFITS OF PIXEL-BASED HIERARCHICAL POLICIES FOR TASK GENERALIZATION

## ABSTRACT

Reinforcement learning practitioners often avoid hierarchical policies, especially in image-based observation spaces. Typically, the single-task performance improvement over flat-policy counterparts does not justify the additional complexity associated with implementing a hierarchy. However, by introducing multiple decision-making levels, hierarchical policies can compose lower-level policies to more effectively generalize between tasks, highlighting the need for multi-task evaluations. We analyze the benefits of hierarchy through simulated multi-task robotic control experiments from pixels. Our results show that hierarchical policies trained with task conditioning can (1) increase performance on training tasks, (2) lead to improved reward and state-space generalizations in similar tasks, and (3) decrease the complexity of fine tuning required to solve novel tasks. Thus, we believe that hierarchical policies should be considered when building reinforcement learning architectures capable of generalizing between tasks.

## 1   INTRODUCTION

General artificial agents aim to solve varied complex tasks and to quickly learn new skills. Two difficulties that typically arise in these settings are long-horizon challenges and generalization to novel environments. The Hierarchical Reinforcement Learning (HRL) framework Dayan & Hinton (1992); Parr & Russell (1997); Sutton et al. (1999) can help address long horizons by discretizing larger tasks into easier sub-tasks. The hierarchy is imposed on the policy: higher levels operate at more abstract time scales and provide sub-goals to lower-level policies, which select the primitive actions. The second problem can be tackled using task conditioning Tenenbaum (1999); Fei-Fei et al. (2003); Rakelly et al. (2019); Zhao et al. (2020). Agents are trained to solve multiple related tasks while receiving characteristics of the current task as input. Developing agents that are able to overcome both aforementioned difficulties is an active area of research.

A recent HRL architecture, called Director Hafner et al. (2022), learns hierarchical behaviors directly from pixels by planning inside the latent space of a learned world model. The high-level policy maximizes both task and exploration rewards by selecting latent goals, while the low-level policy learns to achieve the goals. Despite promising results in some domains, most notably solving mazes from an egocentric view, Director does not outperform baselines in other domains, such as quadruped run in the DeepMind Control Suite (DMC) Tassa et al. (2018). The difference between the successful and unsuccessful domains is primarily in the ability of a successful policy to leverage composition of lower-level behaviors. In locomotion tasks in DMC, the agent needs only to learn to walk, where with maze solving tasks, the agent must learn to walk as well as use that walking ability to navigate.

In addition to single-task composition, a related promise of HRL is transfer of lower-level policies from learned tasks to novel tasks that share a common underlying structure. Recent task conditioned RL work, for example MELD Zhao et al. (2020), has shown that flat (non-hierarchical) policies can learn to generalize. MELD directly encodes the extrinsic reward signal into the latent space model, thus allowing the agent to infer the current task and adapt the policy to it. This method of task conditioning has been successful on training and evaluation tasks with distinct task parameters, which were selected from the same underlying distribution.

In this paper, we explore the effects of hierarchy combined with task conditioning on the performance of RL agents, when presented with novel tasks. We focus on the Director algorithm Hafner et al. (2022) that learns hierarchical policies from pixels, which is depicted in Figure 1.

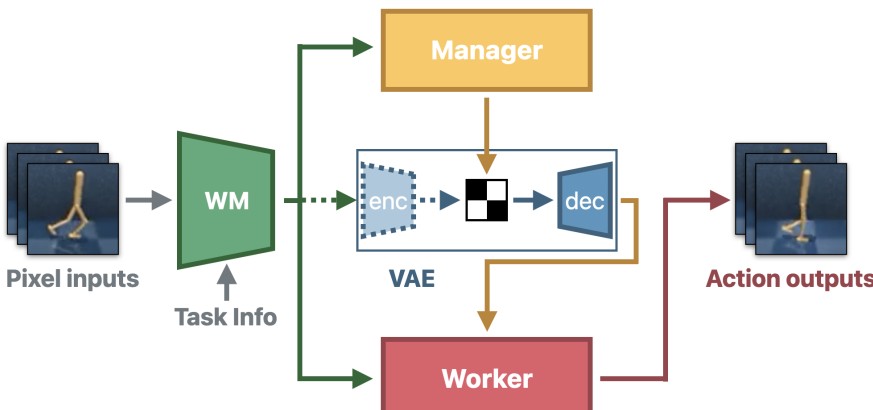

Figure 1: Architecture overview, adapted from Director Hafner et al. (2022), augmented with task conditioning. Image observations and task information, in the form of extrinsic rewards Zhao et al. (2020), are passed to a world model (WM) such as PlaNet Hafner et al. (2019), which encodes them into latent states. These are used to train a categorical VAE and at the same time are passed to the hierarchical policies Hafner et al. (2022), i.e. the higher level policy (called the manager) and the lower level one (called the worker). The manager selects abstract actions in the latent space of the VAE, which are decoded as latent space goal states before passed to the worker. Finally, the worker outputs primitive actions in an attempt to match the goal states set by the manager.

## 2 RELATED WORK

Few previous works showcased convincing results for learning hierarchical policies directly from pixels. Indeed, Director Hafner et al. (2022) was outperformed across a multitude of single tasks by its flat counterpart, DreamerV2 Hafner et al. (2021). For the direct comparisons, see Figures J.1 and K.1 in Hafner et al. (2022).

Other algorithms have shown good performance of hierarchical agents on single tasks when provided with global state information, in the form of XY coordinates of the goal and robot position Nachum et al. (2018) or of a top-down view of the environment Nachum et al. (2019a), when evaluated on quadruped maze tasks.

Previous work on multi-level hierarchical policies in a multi-task setup has typically required training on a diverse distribution of tasks, which were manually specified Frans et al. (2017); Tessler et al. (2017); Veeriah et al. (2021); Rao et al. (2022). Hierarchical Imitation Learning has shown good performance in the multi-task setup Yu et al. (2018); Fox et al. (2019); Gao et al. (2022); Chen et al. (2023), if the expert data quality is high.

Hierarchical structures represent just one approach in tackling compositionality, which is important across other domains as well, such as language modeling and reasoning. Modular computations, based on specialised and reusable sub-NNs, Lepikhin et al. (2021); Fedus et al. (2022) are another approach that have recently shown potential for improving generalization Goyal et al. (2021).

However, the generalizability potential of the hierarchical algorithms mentioned above has remained unclear. To the authors' collective knowledge, the present analysis is the first of its kind dedicated to evaluating the generalizability of task conditioned Hierarchical Reinforcement Learning.

## 3 MOTIVATION

Hierarchical reinforcement learning adds an additional layer of complexity to classical (flat) RL approaches, by grouping sequences of atomic actions into a more structured action space. So what is there to gain from introducing a hierarchy in the action space? We posit that hierarchical RL has three potential advantages, due to the top-down nature of its learning signal: (a) **HRL reduces the effective task horizon**, (b) **HRL**

**learns generalizable and composable skills** and (c) **HRL allows for more sample-efficient few-shot adaptation**.

Consider the task of bipedal locomotion outlined in Figure 2, where a bipedal agent has to navigate towards a goal region. We will use it to showcase the key insights about HRL presented in this work. To evaluate the task conditioned hierarchical agent architecture, we consider a series of multi-task settings.

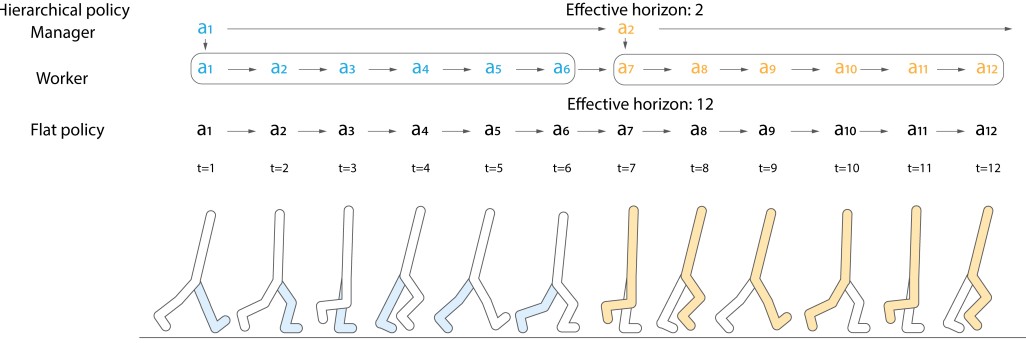

Figure 2: Illustration of the key components of the hierarchical RL paradigm - short effective horizon, compositionality and fast adaptation. The manager prescribes an abstraction action, $a_1$, at time step 1, following which the worker takes primitive actions (joint actuations) for a predetermined period of time (the goal horizon), for example six steps. Only after this sequence of steps has been executed, will the manager take a second action, $a_2$, at time step 7 with the process repeating. Thus, in the figure above, while the flat policy has an effective horizon of 12, the hierarchy has only an effective horizon of 2.

**Shortening the effective task horizon** The bipedal locomotion example outlined in Figure 2 illustrates how decomposing a long horizon problem of length $H$ into $k$ smaller sub-problem of length $H/k$ can accelerate learning Nachum et al. (2019b). Intuitively, the locomotion task require the hierarchical agent to learn strategies for the higher level policy, such as setting goals of different body poses and speeds, while the lower level policy needs to learn high-frequency controls over the joints. Instead of training a single flat policy to produce $H$ actions, it is instead possible to learn a single high-level policy over $H/k$ composite actions (i.e. skills), together with multiple short-horizon low-level policies. Under the assumption that low-level policies are composable and transferable, it is possible to further amortize the training cost by learning the low-level controls on diverse data coming from a multitude of similar tasks, which we present in Section 4.1. To support the claim that the hierarchical structure contributes to performance improvements, rather than an increase in parameters or other confounding cause, we empirically investigate the effects of varying the goal horizon of the higher level policy in Section 4.2. In the limit of short horizons, the hierarchical agent reduces to a flat one, as the higher-level policies outputs new goals for every step of the lower-level one. Similarly, in the limit of long horizons, the hierarchical agent once again reduces to a flat one, as the higher-level policies outputs a single goal for the duration of the entire task, effectively leaving the solution entirely up to lower level policy.

**Zero-shot generalization through compositionality** Figure 2 highlights how the seemingly simple task of bipedal locomotion actually involves a sequence of complex movements. The periodicity of these movements suggests that they could be temporally interchangeable, e.g. the left leg motion could be executed before the motion of the right leg, and vice-versa. Intuitively, composable motions can be recombined in multiple ways, and applied to solve groups of similar tasks - leading to better performance on unseen tasks, i.e. generalization. If the learned low-level policies are indeed composable, they can be used to solve tasks with different reward structures and topologies of the state space. Since the bipedal locomotion tasks provide a limited testbed for our hypotheses, we extend them with an additional challenging domain: quadruped navigation. In this task, a quadruped agent has to reach a goal state based only on RGB visual input. Finding novel goal locations in a maze does not require any low-level locomotion styles beyond what was seen during training. The training results of the navigation tasks are covered in Section 4.1. In Section 4.2 we investigate the agent's reward generalization by requiring the bipedal and quadrupedal walker to move at speeds lower as well as greater than the ones seen in training. Similarly, in Section 4.2, we focus on the agent's state-space generalization by asking it to find target in different maze geometries.

**Fast few-shot adaptation**  Under the premise of compositionality, low-level policies learned during training can then be re-used to solve unseen tasks. This can be done by updating the high-level policy, effectively recombining the learned skills into a new sequence of composite actions. Finetuning the high-level policy requires finding $H/k$ actions, and, since the low-level policies can be frozen (i.e. re-used from training), it is significantly less expensive than finetuning a single flat policy. In Section 4.3, we assess the fine-tuned behavior of a hierarchical agent in the quadruped navigation domain. By freezing various components of the model, we show the dependence of sample complexity on the policy structure - updating low-level policies is less sample-efficient than only updating the high-level policy.

By evaluating with tasks that require transferable and composable worker policies, we highlight the following insights about hierarchical RL:

1. Improved convergence speed on in-domain training tasks (see Section 4.1).

2. Better reward and state-space generalizations in similar evaluation tasks, which share a common underlying structure (see Section 4.2).

3. Decreased complexity of fine-tuning required to solve novel tasks (see Section 4.3).

## 4 EXPERIMENTS

In this study of HRL and flat RL, two categories of tasks are considered: (1) bipedal and quadruped locomotion and (2) quadruped maze navigation.

We select two published agents, which are trained in imagination using actor-critic, share the world model design, have many overlapping NN architecture choices, and their key difference represents that one is a hierarchical policy, Director Hafner et al. (2022), while the other is flat, DreamerV3 Hafner et al. (2023). The default hyper-parameters of these two algorithms are used, apart from increasing the weight associated with the reward loss of the world model. We thus restrict the scope of the paper to comparing HRL and flat RL agents with world models from pixels, but we see no reason why similar analysis could not be performed on other types of HRL and flat RL.

### 4.1 IN-DOMAIN PERFORMANCE

**Locomotion**  The first series of experiments focuses on the locomotion of bipedal and quadrupedal walkers. To provide the agent with task information, we condition on reward. Specifically, we pass the reward as input to the world model, as a way for the agent to infer the task Zhao et al. (2020). For the walker and quadruped walking tasks, the reward is given by $r(s_t, a_t) = 1 - |v(s_t) - v_{\text{target}}|$, which is a measure of how closely the agent manages to match the target speed. $s_t$ is the location of the walker at time step $t$, $v$ is the walker's velocity, $v_{\text{target}}$ is the target velocity, and $a_t$ is the action of the agent at time step $t$.

We evaluate the hierarchical and flat policies, by requiring the policies to train the bipedal and quadruped walkers from DMC suite to walk at various speeds, uniformly sampled between 1.0 and 3.0, and between 0.5 and 1.5 respectively. In DMC suite, the typical bipedal walker tasks are walking (speed 1.0), and running (speed 8.0), while for the quadruped walker they are walking (speed 0.5), and running (speed 5.0). Figure 3 shows the training curves for these task. Notice that both the hierarchical and the flat policy are able to solve the bipedal tasks in a comparable number of steps, however the hierarchy is twice as fast as the flat policy for the quadruped tasks.

When trained on multiple related tasks, the hierarchical agent shows a boost in performance over its flat counterpart. This can likely be attributed to multi-task setting having transferable worker policies across tasks, which can more naturally be leveraged by a hierarchical structure.

**Navigation**  Next, we consider quadruped navigation experiments, specifically the task of reaching a target inside a box. The inputs represent proprioception and egocentric view. We train both the hierarchical and flat policies in a small box, $5 \times 5$, with different colored walls, Figure 4a. The objective is to reach the green ball and whenever this is accomplished, the target is re-spawned somewhere in the area, and the task repeats until the end of the episode. At the start of each training episode, the quadruped begins in the center of the arena with random orientation, however the target initial position is randomly spawned. Note that the task conditioning in this set-up does not depend on the reward and instead is inferred from the pixel inputs, i.e. the image of the target. Nevertheless, in order to speed up the training, the agent is provided

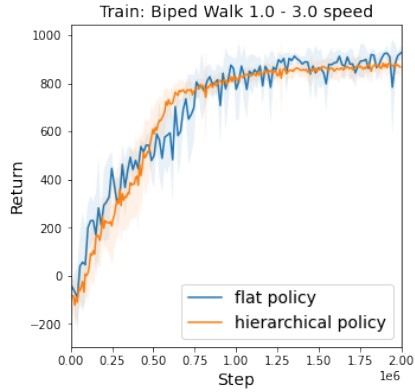 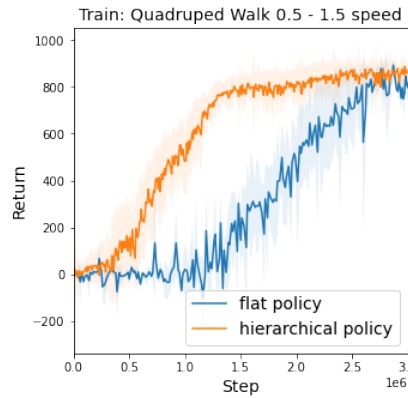

(a) The bipedal training tasks show comparable performance for the hierarchical and flat policies.

(b) The quadruped training tasks show increased performance of the hierarchy over the flat policy.

Figure 3: The training curves for the locomotion tasks.

with a reward of the form: $r(s_t, a_t) = -|s_t - s_{\text{target}}|$, which is a measure of how close the quadruped is to the current target location. $s_{\text{target}}$ is the location of the target.

Figure 4b shows the training curves for this task. Notice that the two policies have similar convergence rates and returns.

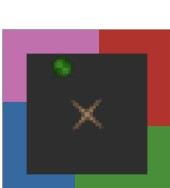 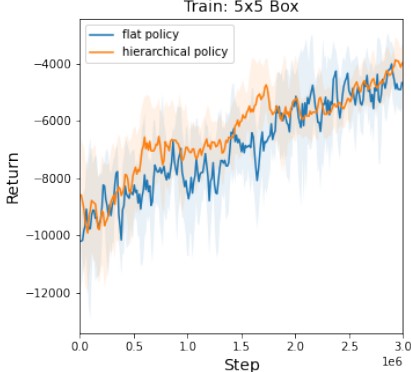

(a) Top view of a $5 \times 5$ box with colored walls, that contains a randomly spawning green sphere target and the quadruped.

(b) The $5 \times 5$ box training task show comparable performance for both the hierarchical and flat policies.

Figure 4: The training task represents a quadruped reaching a re-spawning green spherical target in a $5 \times 5$ box with colored walls.

## 4.2 ZERO-SHOT GENERALIZATION PERFORMANCE

**Locomotion**  Next, we evaluate these trained policies on a broader spectrum of walking speeds, ranging from 0.0 to 8.0 for the bipedal walker, Figure 5, and from 0.0 to 5.0 for the quadruped walker, or in other words from standing still to running. The results are summarised in Figure 6. As expected, both policies are able to move the walker at the target speed similar to those seen in training, thus obtaining high returns. However, for out-of-distribution speeds we notice that the hierarchical policy is able to consistently outperform the flat one.

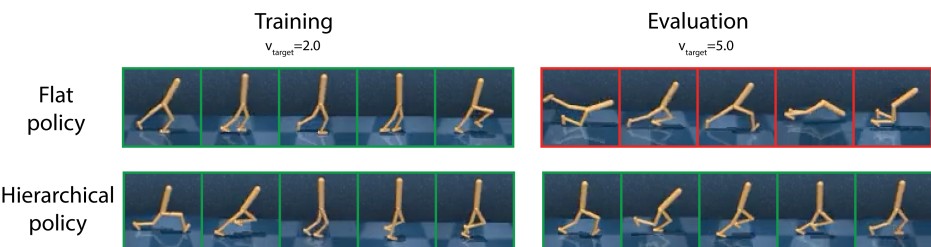

Figure 5: Sequence of image stills from locomotion episodes. While both the hierarchical and the flat policies are able to match the target walking speed under training condition, e.g. $v_{\text{target}} = 2.0$, only the hierarchical agent can generalize to an unseen speed, e.g. $v_{\text{target}} = 5.0$.

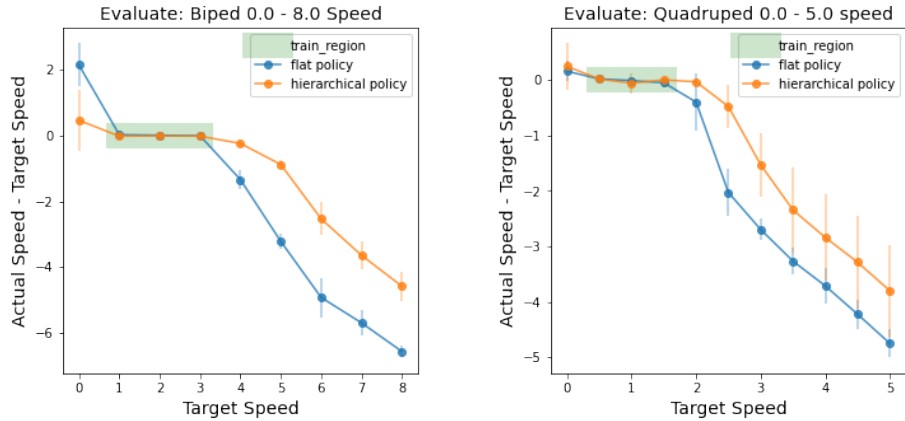

Figure 6: The evaluation curves for the various walking tasks, ranging from standing to running. The bipedal and quadruped out-of-distribution evaluation tasks show that the hierarchical policy outperforms the flat one.

These results show the increased robustness and reward generalizability of the hierarchical policy compared to the flat benchmark, when evaluated on similar tasks that share a common underlying structure. We believe this is due to the higher level policy being able to leverage goals, which the worker is already able to achieve, in order to provide a better policy for the novel task.

**Navigation** During training (Section 4.1), we provide the reward as input to the model to speed up the training and not for task-conditioning purposes. During evaluation, to ensure that the reward is not task-conditioning and that the quadruped does not just follow this guiding reward to the target, we change the reward function. Unless the quadruped touches the target, i.e. $\mathbb{I}[||s_t - s_{\text{target}}||_2 < \varepsilon]$, the system receives a large and constant negative reward (which in our case is $-8.0$), regardless of the distance between the quadruped and the target. Hence, the reward cannot guide the quadruped or condition the task. If the quadruped touches the target, the agent is rewarded with the negative radius of the target (which in our case is $0.5$). Nevertheless, changing the reward formulation could negatively impact the performance of an agent, if that agent has learned to blindly follow the guiding reward instead of looking for the target and then moving towards its.

We start by reporting the evaluation results for the $5 \times 5$ arena that was also used in training. For an episode of set length of 3000 steps, the hierarchical policy manages to reach the re-spawning target an average of 29.5 times per episode, while the flat one obtains an average of 12. The results are summarised in Table 1 column 2. This difference might be surprising, since Figure 4b shows comparable training curves. One can attribute this to the constant reward signal used at evaluation, which can no longer serve as a guide for the agent. It suggests that the hierarchical policy has learned to focus more on the pixel inputs than the flat policy has, in order to reach the target.

Next, we evaluate the agents in three increasingly more complex environments, shown in Figure 7. In the $9 \times 9$ box, Figure 7a, the quadruped starts off in the center of the arena with random orientation, and the goal is

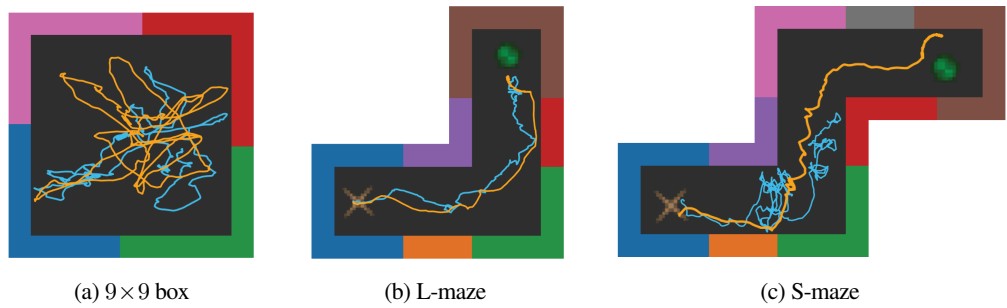

| (a) $9 \times 9$ box | (b) L-maze | (c) S-maze |

Figure 7: Evaluation tasks in order of increasing topological complexity. Sample evaluation trajectories from both the hierarchical agent and the flat agent. (a) contains a randomly spawning green sphere target. Same color walls as the $5 \times 5$ box. (b) contains a stationary green sphere target and a corner. Increased color variation of walls as compared to the $9 \times 9$ box. (c) contains a stationary green sphere target and two corners. Increased color variation of walls as compared to the L-shaped maze.

| Agent | $5 \times 5$ Box | $9 \times 9$ Box | L-maze | S-maze |
|---|---|---|---|---|
| flat policy | 12 | 3 | 90% (1.5K steps) | 0% |
| hierarchical policy | 29.5 | 14 | 100% (0.8K steps) | 60% (1.2K steps) |

Table 1: Comparing the performance of the hierarchical and flat policies on the locomotion tasks. The $5 \times 5$ and $9 \times 9$ box tasks have re-spawning targets. Columns 2 and 3 represent the number of targets reached within an episode. The L-maze and S-maze tasks have fixed targets. Columns 4 and 5 represent the task success percentage and the number of steps agent requires to reach the target.

randomly spawned. For a 3000 step episode, the hierarchical policy manages to reach the re-spawning target an average of 14 times per episode, while the flat one only obtains an average of 3 (see Table 1 column 3).

Figures 7b and 7c further complicate the task by introducing corners and extra colors on the walls. The targets are stationary and placed furthest away from the quadruped, which in turn is always spawned in a fixed starting location, but with random orientation. The hierarchical policy has a 100% success rate on the L-maze in Figure 7b, and 60% on the S-maze in Figure 7c. The flat policy has a 90% success rate in the L-maze, but it takes on average twice the number of steps to reach the goal compared to the hierarchical agent. The flat policy is unable to solve the S-maze. Table 1 columns 4 and 5 encapsulate these results. Also, the sample trajectories shown in Figure 7 indicate smoother trajectories in general for the hierarchical agent compared to the flat one.

Hence, for similar tasks, which share a common underlying structure, the hierarchy leads to improved state-space generalizations, in addition to the reward generalization previously described.

**Ablation of goal selection frequency**  A key hyper-parameter of the hierarchical structure is the goal horizon length of the higher-level policy. That is the duration for which a goal is kept constant and the lower-level policy is attempting to reach it. This horizon defines the scale of the temporal abstraction. Throughout our analyses, we have used the same default horizon length as in the Director paper, specifically 8. As the flat policy benchmark has an action repeat of two, while the hierarchical one has an action repeat of one, the former resembles most closely the hierarchy with a goal horizon of length two. Figure 8 shows the effects of varying the goal horizon length for the bipedal and quadruped walking tasks. The hierarchical structure increases task generalizability compared to the flat policy, regardless of goal horizon length. Furthermore, there appears to be an optimal value for said length, which for the present set-up is 8. Moving away from this values leads to varying decreases in performance.

### 4.3 FEW-SHOT GENERALIZATION PERFORMANCE

We investigate the adaptability of a pre-trained hierarchy via fine-tuning subsets of the policy's components. We choose the quadruped maze as a fine-tuning task, since the learned lower level policies are sufficient

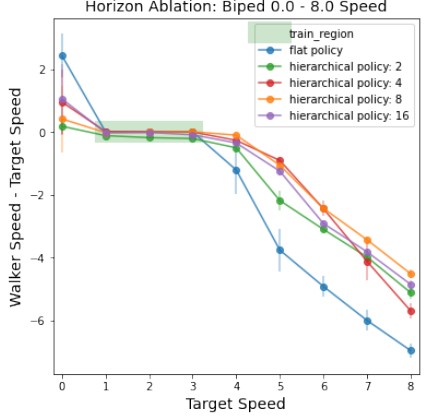

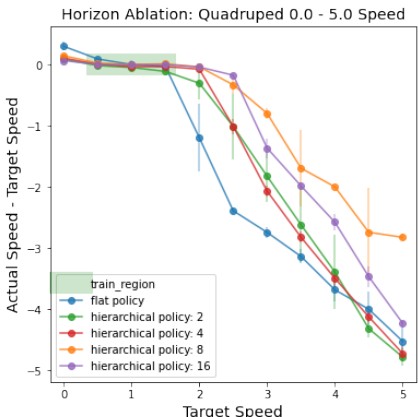

(a) Absolute error of the walking speed task for the bipedal walker tasks show that the hierarchical policy outperforms the flat one.

(b) The quadruped out-of-distribution evaluation tasks show that the hierarchical policy outperforms the flat one.

Figure 8: Evaluation curves for the bipedal and quadruped walking tasks, with target speeds ranging from standing to running, for various goal horizons. In general, the hierarchical policies outperform the flat benchmarks and there is an optimal value for the horizon length. In the case of our architecture set-up, that value is 8.

to move the agent reliably. Specifically, the lower level policies have already learned how to walk and achieve goals set by higher level ones.

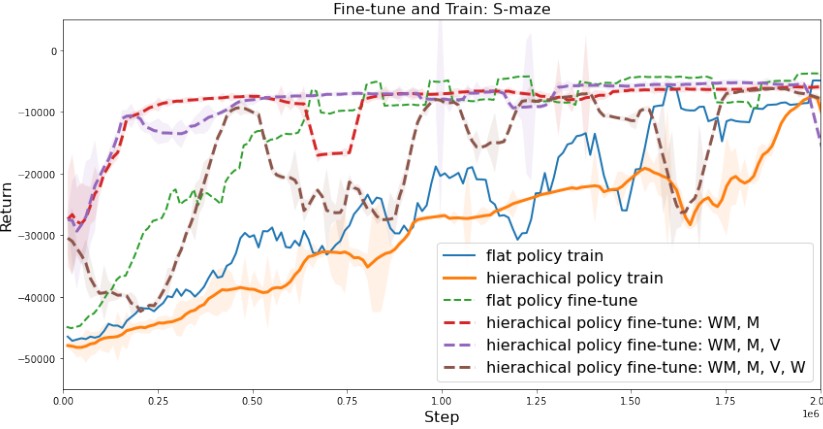

Figure 9: Fine-tuning and training curves on the S-shape maze. The various components of the hierarchy are the world model (WM), the higher-level manager (M), the goal VAE (V), and the lower-level worker (W). At the -10,000 return mark, the task is effectively solved with 100% episode success rate. Fine-tuning only the WM and the Manager is needed to solve the task, which has higher convergence speed compared to fine-tuning the entire hierarchy, fine-tuning the flat policy, and training from scratch either the hierarchical agent or the flat one.

We focus on the S-shaped maze, Figure 7c, as it represents a more complex task, for which the hierarchical policy was not very effective at performing zero-shot learning. Figure9 shows the return as a function of training step for a variety of fine-tuning settings. We observe that fine-tuning only the higher level policy and the world model is sufficient to achieve success in a rapid and stable manner. Fine-tuning the world model is required as the task presents new geometries, such as corners, and a doubling in the number of wall colors. Fine-tuning the manager allows the higher-level policies to set appropriate goals in this

new environment. Notice that no fine-tuning of the lower level policy is necessary as the agent does not need to relearn how to walk.

While fine-tuning the entire algorithm will produce an agent capable of efficiently solving the task, the convergence is much slower compared to only fine-tuning the world model and the higher-level policy. Fine-tuning the pre-trained flat policy will also result in an agent that is able to solve the task, but it is slower to convergence compared to the hierarchical agent. Finally, one could also train either agent (the hierarchical or the flat one) on the task from scratch, but convergence would be an order of magnitude slower than just fine-tuning the higher-level policy and world model. Table 2 provides the steps required to converge.

| Flat | | Hierarchical | | | |
| --- | --- | --- | --- | --- | --- |
| scratch | fine-tune | scratch | WM & M | WM & M & V | WM & M & V & W |
| 1,560 | 660 | 1,920 | 220 | 430 | 960 |

Table 2: Number of steps (in thousands) required for solving the S-maze task, defined as crossing the -10,000 return mark, for fine-tuning or training from scratch. For Hierarchical fine tuning, we indicate what parts of the model were fine-tuned.

Because of their compositionality, hierarchical policies decrease the complexity of fine-tuning required to solve novel tasks compared to the flat policies.

## 5 DISCUSSION

One limitation of the original Director Hafner et al. (2022) paper was the evaluation of the algorithm. Across most of the DMC suite tasks, it was outperformed by a flat policy approach Hafner et al. (2021). The benefits of the hierarchy were shown when training and evaluating on single maze tasks, in particular when the geometry was complex (for example multiple corners and increased distance to the target). These suggested that hierarchical agents benefit from tasks that display compositionality, yet this angle was not discussed and further analysis was needed to better understand it.

We are able to extend the task conditioning paradigm to the HRL architecture and benchmark it against a flat policy. In the process, we uncovered that, while flat policies have been known to learn and adapt to new in-distribution task parameters Zhao et al. (2020), they tend to rapidly falter as these parameters extend out-of distribution. Our findings show that hierarchical policies can not only increase performance on training tasks, but also lead to improved reward and state-space generalizations in similar tasks, allowing agents to solve them with a higher degree of success. Furthermore, hierarchical policies can decrease the complexity of fine-tuning required to solve novel tasks, by only focusing on training the higher level policies.

Future work is needed to continue the exploration of the generalization potential of hierarchical structures. One such direction would be improving the goal conditioning of the worker, possibly using a LLM, so that it learns abstract commands, e.g. go left, go forward, as opposed to learning to match a specific latent state. This would allow agents to further abstract details of the environment that are not shared across similar tasks (for example changing the color of the floor for the walker or quadruped between training and evaluation tasks).

Hence, we believe it worthwhile to further pursue research into hierarchical reinforcement architectures and invite practitioners to consider hierarchy when requiring agents capable of generalization.

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

## A    APPENDIX

The associated code-base will be released upon acceptance of the present manuscript.

