# OpenReview forum: "On the benefits of pixel-based hierarchical policies for task generalization"
_ICLR.cc/2024/Conference — ICLR 2024 Conference Withdrawn Submission_

### Official Review · Reviewer_eEP2 · 2023-10-31

**Soundness:** 1 poor
**Presentation:** 1 poor
**Contribution:** 1 poor
**Rating:** 1
**Confidence:** 4

**Summary:**

This is a study paper evaluating benefits of hierarchical policies over flat policies when training RL based agents. It shows several benefits of a hierarchical method (Director) over a flat method (Dreamer) across a series of tasks.

**Strengths:**

The authors study an important problem: using hierarchical RL policies for robotic control. While the benefits are clear, but practically there is little evidence in general that they are better than flat. They include several experiments which demonstrate the effectiveness of HRL methods over flat policies.

**Weaknesses:**

There are significant writing and basic setup issues in this paper. At a high level, the paper is a study comparing hierarchical and flat policies but, it is extremely limited in scope: it evaluates exactly one method per category (hierarchical and flat and that too only model-based methods. Furthermore, the work only evaluates on a single class of tasks (locomotion). This sort of evaluation is not nearly comprehensive enough for a study paper.

There are also very clear writing/claiming issues. Statements such as "We posit that hierarchical RL has three potential advantages,
due to the top-down nature of its learning signal: (a) HRL reduces the effective task horizon, (b) HRL learns generalizable and composable skills and (c) HRL allows for more sample-efficient few-shot adaptation." makes it appear that the authors are the first to consider the advantages of hierarchical RL. There is a wide body of work on hierarchical RL that has considered and noted similar advantages. While there are many relevant papers, I refer the authors to HIRO (Nachum et al) and Latent Skill Planning (Xie et al) for a recent treatment as well as the options framework (Sutton et all 1999) for more background. There is very little in the way of citation, this work seems largely incomplete.

**Questions:**

Please add additional experiments across significantly more domains: Metaworld, Robosuite, DMC, Maniskill, many more hierarchical RL algorithms (such as HIRO) which are more representative and flesh out the writing significantly.

---

### Official Review · Reviewer_guTM · 2023-11-01

**Soundness:** 1 poor
**Presentation:** 2 fair
**Contribution:** 1 poor
**Rating:** 3
**Confidence:** 3

**Summary:**

The paper presents some empirical results comparison of Hierarchical (model-based) RL and regular RL with a focus on three insights: 1. HRL increases
performance on training tasks. 2. HRL leads to improved reward and state-space generalizations in similar tasks. (zero-shot generalization) 3. HRL decrease the complexity of fine tuning required to solve novel tasks. (few-shot generalization)

**Strengths:**

1. The authors provide some interesting empirical results for model-based HRL methods.
2. The paper is generally easy to follow.

**Weaknesses:**

As the main contribution of the paper seems to focus on the empirical comparison of existing paper, I think 1. More domains should be tested. There are only two domains considered in the experiments and both of them are designed by the authors. I suggest the authors evaluate on some HRL benchmarks. 2. HRL in the model-free direction has been explored a lot so if the authors want to draw a conclusion about general HRL, the model free methods should also be analyzed in terms of few/zero-shot adaptation. E.g. [1]

Section 4.1, it is not clear to me what the author's conclusion is given the experimental results. It seems that two existing world-model style methods are compared and the hierarchical one is better than the flat policy one in only one domain. I don't think there is enough empirical evidence draw a conclusion.

Section 4.2, it is not clear to me what the green train region means in the plots. Besides, I suggest the authors save the plots in the form of .pdf and put them in the paper as the plots in the current draft are a little blurry,

Section 4.3, the authors should explain in more detail about what  WM, M, and V are. Otherwise it's hard to understand what the results imply. Also, the authors should provide the information about how many random seeds are run for each baseline.

The insights provided in the motivation section are not new. It has been proved that hierarchical RL structure benefits the transfer and adaptation of RL agent. E.g., see [2].

The code or the hyperparameter settings of the experiments are not provided. Details of the environments are also not provided.

[1] Data-Efficient Hierarchical Reinforcement Learning. Neurips 2018.

[2] Meta-learning parameterized skills. ICML 2023.

**Questions:**

See Weaknesses.

---

### Official Review · Reviewer_yc95 · 2023-11-01

**Soundness:** 3 good
**Presentation:** 2 fair
**Contribution:** 1 poor
**Rating:** 3
**Confidence:** 3

**Summary:**

The paper discusses the benefits of pixel-based hierarchical policies for task generalization. Specifically, it explores the effects of hierarchy combined with task conditioning ( Zhao et al. (2020)) on the performance of RL agents, when presented with novel tasks. It focuses on the Director algorithm (Hafner et al. (2022)) that learns hierarchical policies from pixels.

The primary findings from multi-task robotic control experiments suggest that hierarchical policies trained with task conditioning can

1. increase performance on training tasks
2. lead to improved reward and state-space generalizations in similar tasks
3. decrease the complexity of fine-tuning required to solve novel tasks.

**Strengths:**

The strengths of this paper include:

- The authors comprehensively analyze the potential advantages of hierarchical RL, including reduced effective task horizon, generalizable and composable skills, and sample-efficient few-shot adaptation. This  provides readers with a holistic understanding.
- The experiments are conducted on two different two categories of tasks (bipedal / quadruped locomotion and quadruped maze navigation.), so the conclusions will not be too biased.

**Weaknesses:**

The weaknesses of this paper include:

- This paper conducts all experiments based on Director (Hafner et al. (2022)), so only one backbone hierarchical RL method is considered. Relying solely on one hierarchical RL method can narrow the scope of the study. There might be other methods or variations in the field that offer different perspectives or results. By not considering multiple backbones or approaches, the paper might miss out on capturing a broader and more diverse set of insights.

- The effect of task conditioning is not clearly explained. I assume there are some reasons to combine task conditioning into hierarchical policies. If its effects are not lucidly explained, readers might struggle to understand its significance, implementation, and outcomes in the presented method.

- This paper conducts an empirical study instead of proposing a novel method. While empirical studies are valuable, the conclusions from this paper are not too surprising.

- The presentation of this paper can be improved. In particular,
    - The figures should be polished.
    - The motivation of each experiment should be explained more clearly.
    - The contributions of the paper should be explicitly stated.

**Questions:**

N/A

---

### Official Review · Reviewer_TyzN · 2023-11-10

**Soundness:** 2 fair
**Presentation:** 3 good
**Contribution:** 2 fair
**Rating:** 3
**Confidence:** 4

**Summary:**

This paper aims to understand the effectiveness of Hierarchical reinforcement learning (HRL) for multi-task settings. The specific scope of this paper is cases with visual observations. The motivation for studying this problem is that hierarchical policies, by construction, learn some low-level skills/policies which might be useful when transferring to new environments or when dealing with a host of different settings/environments/tasks. Other benefits include the computational efficiency of HRL (due to a shorter horizon). The hierarchical framework explored Director (Hafner et al., 2022), which operates in latent action space, later decoded into low-level actions. The main addition to this framework is a task conditioning input passed to the world model of Director.  The experiments are conducted on two sets of tasks: bipedal and quadruped locomotion as well as quadruped navigation. The experiments show that HRL policies perform roughly the same as flat policies during training. However, when the environment or task is changed slightly (like the size/shape of the box changes for navigation or the speed required to walk at changes for locomotion), the performance drop in a flat policy is much more. The same holds for adaptation to the new task or environment - HRL policies adapt faster.

**Strengths:**

- To my knowledge this type of study of HRL is novel
- I think this is an important direction, as task generalization has the potential for a big impact, especially when deploying in the real world
- The paper is well written and easy to follow
- The use of HRL is well motivated and the problem statement is clear
- I appreciate the detailed analysis of the experiments/ablations for different types of generalization

**Weaknesses:**

- There are many different ways to do task adaptation that do not require HRL, I think those should also be studied in this paper.

- I would like to see this done on more complicated settings, as locomotion does not require much visual knowledge. Manipulation tasks could be interesting for this. Open-world settings like Minecraft would also be a good test for some of these experiments.

- More drastic environment and task changes are also important - I don't think changing the target speed requires that much adaptation (maybe one could go from walking to jumping or climbing stairs).

**Questions:**

See weaknesses